

# Positive effect of dietary lutein and cholesterol on the undirected song activity of an opportunistic breeder

Stefania Casagrande[1,2], Rianne Pinxten[2,3], Erika Zaid[2,4] and Marcel Eens[2]

[1] Evolutionary Physiology Group, Max Planck Institute for Ornithology, Seewiesen, Germany
[2] Department of Biology, Behavioural Ecology and Ecophysiology Group, University of Antwerp, Antwerp, Belgium
[3] Faculty of Social Sciences, Antwerp School of Education, University of Antwerp, Antwerp, Belgium
[4] Department of Zoology, School of Life Sciences, La Trobe University, Melbourne, Victoria, Australia

Corresponding author
Stefania Casagrande,
scasagrande@orn.mpg.de

## ABSTRACT

Song is a sexually selected trait that is thought to be an honest signal of the health condition of an individual in many bird species. For species that breed opportunistically, the quantity of food may be a determinant of singing activity. However, it is not yet known whether the quality of food plays an important role in this respect. The aim of the present study was to experimentally investigate the role of two calorie-free nutrients (lutein and cholesterol) in determining the expression of a sexually selected behavior (song rate) and other behaviors (locomotor activity, self-maintenance activity, eating and resting) in male zebra finches (*Taeniopygia guttata*). We predicted that males supplemented with lutein and cholesterol would sing at higher rates than controls because both lutein and cholesterol have important health-related physiological functions in birds and birdsong mirrors individual condition. To control for testosterone secretion that may upregulate birdsong, birds were exposed to a decreasing photoperiod. Our results showed that control males down-regulated testosterone in response to a decreasing photoperiod, while birds treated with lutein or cholesterol maintained a constant singing activity. Both lutein- and cholesterol-supplemented groups sang more than control groups by the end of the experiment, indicating that the quality of food can affect undirected song irrespective of circulating testosterone concentrations. None of the other measured behaviors were affected by the treatment, suggesting that, when individuals have full availability of food, sexually selected song traits are more sensitive to the effect of food quality than other behavioral traits. Overall the results support our prediction that undirected song produced by male zebra finches signals access to high-quality food.

## INTRODUCTION

Individuals of different species have evolved a wide array of traits to signal their condition to perspective potential mates or rivals (*Von Schantz et al., 1999*; *Johnstone, Rands & Evans, 2009*; *Hill, 2011*). The communicative potential of songbirds relies, in large part, on the production of complex and prolonged songs that are evaluated by conspecifics

to assess the "quality" (*sensu Wilson & Nussey, 2010*) of the emitter (*Duffy & Ball, 2002*; *Garamszegi, Møller & Erritzøe, 2003*; *Møller, Henry & Erritzøe, 2000*; *Nowicki & Searcy, 2004*; *Tregenza et al., 2006*). Song can be an expression of quality because its production is a finely-tuned process easily limited by several factors (reviewed in *Gil & Gahr, 2002*). Although the matter is still debated (e.g., *Ophir, Schrader & Gillooly, 2010*; *Ward, 2004*), the existence of direct costs sustained by the increase of oxygen consumption associated with singing activity, suggests that it is an energetically demanding task (*Franz & Goller, 2003*; *Hasselquist & Bensch, 2008*; *Zollinger, Goller & Brumm, 2011*; but see *Ward, 2004*), which must be traded-off with other activities. For example, it has been observed that birdsong decreases along with lowering temperatures, probably resulting from an increased need to maintain thermostasis (*Dunn & Zann, 1996*). Even more convincing is the evidence from the link between food availability and singing. For example, male pied flycatchers (*Ficedula hypoleuca*) provided with supplemental food sang more and had more complex songs than controls (*Lampe & Espmark, 2002*). Similarly, dunnock males (*Prunella modularis*) supplemented with additional food sang at higher rates than control males (*Davies & Lundberg, 1984*). Other species, such as the European starling (*Sturnus vulgaris*) sing at higher rates when they have a better nutritional status as suggested by high levels of cholesterol and albumin in the plasma (*Van Hout et al., 2012*). The association between the nutritional state of singing birds and their song performance can be explained by the activation of the metabolic pathways that are needed to exhibit an energy demanding activity such as singing (*Hasselquist & Bensch, 2008*). On the other side, food shortage can have a negative impact on birdsong. In the zebra finch (*Taeniopygia guttata*), *Lynn et al. (2010)* found a reduction in the expression of undirected song and courtship behavior following 4 h of fasting. Similarly, *Ritschard & Brumm (2012)* found that zebra finches kept under food restriction performed a lower rate of undirected song than controls. Another study reported that this relationship was mediated by male condition (indicated by body mass; *Geberzahn & Gahr, 2011*). Despite the existence of a clear association between the nutritional state and song performance, studies on how food quality can affect song remain very rare (but see *Casagrande et al., 2015*; *Casagrande et al., 2014*; *Van Hout, Eens & Pinxten, 2011*).

In general, males of almost all songbird species produce song in response to the presence of a conspecific with the intent of attracting it (courtship) or repelling it (agonistic interactions), or more generally, to induce a modification in the behavior of the listener (*Riters, 2011*). In zebra finches this type of song is named "directed song," which is usually regulated by sex steroids (*Catchpole & Slater, 2008*) and produced in contexts related to reproduction (*Riebel, 2009*). In addition to directed song, zebra finches also sing an "undirected song," which is not directed at a particular individual. Directed and undirected songs also differ bioacoustically in the rate of syllable delivery, stereotypy and number of introductory elements (*Riebel, 2009*), as well as in their neural, genetic and endocrinological control (*Jarvis et al., 1998*). Although sexual selection probably acts less intensely on undirected song compared to directed song, it can be considered a sexually selected trait because it can enhance the chance of mating by attracting a mate when the female of the social pair is not present, gaining a higher chance of re-mating in case of partner loss, maintaining pair bonds and increasing the frequency of extra-pair copulations

(*Dunn & Zann, 1996*; *Dunn & Zann, 1997*). Laboratory experiments show that undirected song is indeed attractive to female zebra finches (*Holveck & Riebel, 2007*; *Jesse & Riebel, 2012*; *Tomaszycki & Adkins-Regan, 2005*). Studies carried out in free-living zebra finches (*Taeniopygia guttata*) revealed that undirected song is performed at high levels throughout the year and that it is more frequent than directed song with clear signaling functions even related to sexual behavior (*Dunn & Zann, 1996*). For example, higher rates of undirected song performed near the nest correlate with the presence of the females in the nest, suggesting a within-pair function of undirected song (*Dunn & Zann, 1996*). Even though undirected song can also be produced independently of sex steroids (*Pröve, 1974*) or an active reproductive state (*Perfito et al., 2008*), it can respond to androgens similarly to directed song (*Pröve & Immelmann, 1982*; *Walters, Collado & Harding, 1991*), but may have a much lower threshold for hormonal activation (*Pröve, 1974*).

It has now been repeatedly shown that dietary lutein, which is a calorie-free micronutrient, promotes the undirected song duration of male European starlings (*Casagrande et al., 2014*; *Van Hout, Eens & Pinxten, 2011*), even in males that were coping with an inflammatory response (*Casagrande et al., 2015*). Lutein is a hydroxy-carotenoid synthetized by autotrophic organisms that, besides acting as a pigment and as an enhancer of the immune system (*Svensson & Wong, 2011*), is known mostly for its antioxidant properties (*Hill & Johnson, 2012*; *Vinkler & Albrecht, 2010*; *Pérez-Rodríguez, 2009*; *Casagrande et al., 2014*). As with all carotenoids, lutein cannot be synthetized by animals and thus represents a potentially limited resource obtained only by eating lutein-rich food. Consequently, it is generally expected that, since lutein is a limited resource with beneficial effects on the oxidative status, and since birdsong can mirror oxidative condition (*Casagrande, Pinxten & Eens, 2016*), birds with access to lutein-rich food can potentially improve their singing performance and, consequently, fitness outcome.

Another multi-functional molecule that can be acquired through the diet, but that also can be synthesized endogenously is cholesterol, a calorie-free sterol which is known to determine the structure and functionality of the animal cells' membrane. It has also been demonstrated that within certain concentrations cholesterol has clear antioxidant functions that improve the health status of vertebrates (*Brown & Galea, 2010*; *Casagrande et al., 2014*; *Murphy & Johnson, 2008*; *Smith, 1991*). Recently, a relationship has also been reported between cholesterol and the expression of song in European starlings, where circulating cholesterol levels were positively associated with song rate (*Van Hout et al., 2012*).

The general aim of the present study was to experimentally investigate the effect of these calorie-free micronutrients on the duration of undirected song. Specifically, our study aimed at determining whether a potentially ecologically relevant cue such as food quality can affect undirected singing behavior.

We addressed this question in the zebra finch, a granivorous, non-territorial and colonial species adapted to breed opportunistically (*Zann, 1996*), but that in more temperate regions of its range (Australia and Indonesia) is more sensitive to photoperiodic fluctuations (*Zann, 1996*; *Zann et al., 1995*). This opportunistic species has evolved in unpredictable environments and is highly responsive to local non-seasonal conditions such as water and food availability (*Perfito, Bentley & Hau, 2006*; *Perfito et al., 2008*; *Prior & Soma, 2015*).

It has also been recently discovered that the structure of the neural song system shows a seasonal-like variation that may be related to the reproductive state (testis volume), although investigations of the relationship between song nuclei and circulating sex steroids are still missing (*Perfito et al., 2015*).

Birds were kept under a dynamically decreasing photoperiod, as sex steroids are predicted to decrease with the shortening of daylight also in this opportunistic species (*Bentley et al., 2000*; *Perfito et al., 2008*). We determined the hormonal profile of individuals by measuring circulating testosterone. Since this hormone can regulate a wide array of behavioral patterns related to the breeding state, and since sexually selected and naturally selected traits can have different sensitivity to both hormones and micronutrients (*Cotton, Fowler & Pomiankowski, 2004*; *Johnstone, Rands & Evans, 2009*; *Van Hout, Eens & Pinxten, 2011*), we recorded the effect of our treatments on singing behavior, as well as on other observed behaviors (see below). We expected that: 1. Birds would decrease undirected song when T concentrations decrease when kept on a standard diet; 2. Lutein and cholesterol would positively affect singing activity, but not other behaviors, because the positive effect of micronutrients will be more pronounced on a sexual trait such as undirected song.

## MATERIALS & METHODS

### Experimental design and food manipulation

The study was approved by the ethical committee for animal experiments (ECD) of the University of Antwerp (ID number: 2014-21). Forty-three adult zebra finch males of similar age were acquired from a local authorized pet shop with certified origin. Birds were allowed to acclimatize for three weeks before the onset of the experiment. Each male was housed in an individual cage (L × W × H: 60 × 25 × 35 cm) equipped with 3 perches, birds' grit and *ad libitum* access to food, water and cuttlefish bone. Extra water for bathing was offered on a weekly basis to allow the birds to accomplish natural self-maintenance activities. The cages were adjacent to each other, arranged against two opposite walls of an animal-approved indoor room at the campus Drie Eiken of the University of Antwerp, where birds (only males were present in the room) were in constant visual and auditory contact with each other. Males of different treatments were randomly positioned in one of the 43 cages in order to avoid any bias imputable to the location of the cage. Birds were exposed to constant temperature (about 20 °C) and short decreasing photoperiod resembling the photoperiod variation observed in Antwerp during the course of the experiment (variation in duration of daylight between 6 November and 6 December 2012: 9 h 22 m–8 h 04 m). The light was switched on and off by an automatic system and included a gradual change of light to mimic twilight that lasted 30 min after 'lights on' and 30 min before 'lights off'. The modification of the day length was performed manually daily by the researchers. All birds, irrespective of the treatment, were fed an *ad libitum* diet of water and a mix of seeds for finches containing about 2.75 $\mu g^* g^{-1}$ of lutein and zeaxanthin (respectively 10:1) (*Casagrande et al., 2011*) and free of cholesterol. Since zebra finches usually consume at least about 3 g of seeds/day (*Perfito et al., 2008*), the baseline average amount of lutein intake was estimated to be at least 8 $\mu g$/day.

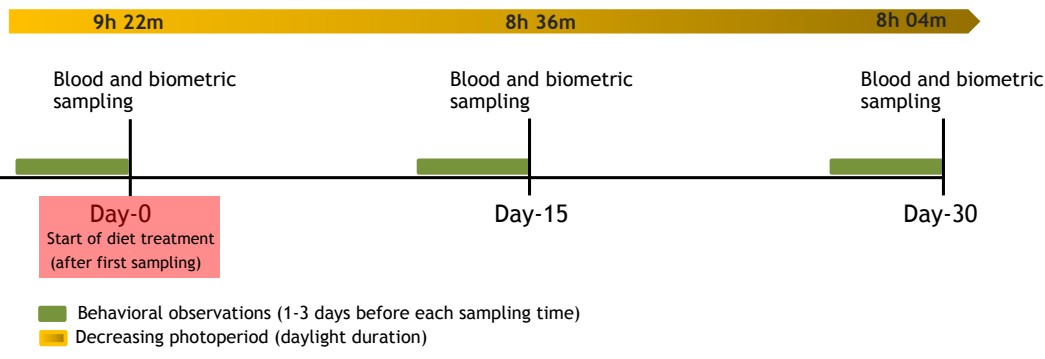

**Figure 1  Timing of the experiment.**

On day 0, three different groups were created (Controls, Caro-males and Chol-males) for the four weeks of the experiment (see Fig. 1 for the timing of the experiment). Fourteen randomly selected males (Caro-males) received an additional 15 µg*ml$^{-1}$ of lutein (ORO GLO$^{TM}$; Kemin Industries Inc., Des Moines, Iowa, USA; extracted from marigolds *Tagetes erecta*) in drinking water provided in an opaque dispenser to prevent photo-oxidation (*Alonso-Alvarez et al., 2004*; *Blount et al., 2003*; *McGraw & Ardia, 2003*). Considering a daily average intake of 2 ml of water per zebra finch fed the baseline seed diet of the studied colony and of other colonies kept under the same conditions (*McGraw & Ardia, 2003*), the daily dose of lutein assumed during the experiment by each individual was 38 µg/day, approximately five times the amount of lutein eaten by the 14 controls birds (C birds). Other 15 males (Chol-males) were fed seeds enriched with 2% powdered cholesterol (Product code: 14606-100G-F; Sigma-Aldrich) following the protocol described by previous studies in the same species (*Allen & Wong, 1993*; *McGraw & Parker, 2006*). Seeds provided to C and Caro birds were cholesterol free. Based on a previous study, it is known that the provided concentration nearly doubles blood cholesterol levels in zebra finches over the course of 3–4 weeks (*McGraw & Parker, 2006*).

## Behavioral observations and analysis of recordings

We video-recorded the behavior of birds for 4 h starting one hour after morning twilight, one to three days before each sampling day (day 0, day 15 and day 30, respectively; see also Fig. 1). In total 14–15 birds were simultaneously recorded each day, assuring a balanced representation of the 3 treatments. The recordings were subsequently analyzed in random order by a single person (EZ) using a software for the analysis of behavioral data (The Observer XT; Noldus Information Technologies, Wageningen, The Netherlands). In order to obtain an accurate measure of all behaviors, even the ones that lasted for a very short time, the software was set to automatically determine the duration of each behavior (event recorder) while the observer was watching the video and hearing the sound. All activities of the focal bird were analyzed using a segment of 20 min of recording, randomly chosen within the 4 h of recording to avoid any potential bias due to timing of activities. This amount of time was approximately twice the duration that is usually used to assess activity levels in this species (*Lopes, Wingfield & Bentley, 2012*). We discarded the first 30 min of

recordings as during this period the behavior may still have been potentially affected by the presence of the researchers that had installed the camera. The activities were quantified as the total duration (in seconds) of resting (perching and standing on the ground), self-maintenance activities (preening, scratching, and beak rubbing), locomotor activity (hopping and flying), eating and singing undirected song, respectively. Although directed and undirected songs are structurally very similar, undirected song is, by definition, not directed toward a female and it is not associated with specific behavioral patterns related to the breeding condition (*Sossinka & Boehner, 1980*).

On day 0 (start of treatment, first measure before food manipulation), day 15 and day 30 birds were weighed to the nearest gram to examine the effect of dietary treatments on the body mass. Just before the measurement, males were blood sampled by venipuncturing the brachial vein collecting 80 µL of blood with heparinized microcapillary tubes. Plasma was removed from centrifuged blood and stored at −80 °C in 1.5 mL Eppendorf tubes for T and lutein analyses (four months later).

### Hormonal assay

T concentrations were determined using enzyme immonoessay (EIA) kits (Cat. No. ADI-901-065; Enzo Life Sciences, Farmingdale, New York, USA) following a diethyl ether extraction of 25 µL sample volume. After drying the extract under N2 stream, 250 µL of Assay Buffer was added (1:10 dilution), and the samples were allowed to reconstitute overnight at 4 °C. A stripped plasma sample spiked with a known amount of T (2 ng*mL$^{-1}$) as well as one blank sample containing only assay buffer were taken through the entire assay procedure. The next day, 100 µL of each sample (in duplicate) was added to individual wells on the assay plate alongside a standard curve with 5 points ranging from 7.81 pg to 2,000.00 pg*mL$^{-1}$. The samples were distributed randomly within and across plates but an individual's repeated samples were always included on the same plate. The plate was read on a microplate reader (VersaMax; Molecular Devices Inc., Sunnyvale, California, USA) at 405 nm with a correction wavelength set at 570 nm. The average extraction efficiency was 70% and final values were corrected accordingly. The average lower sensitivity of the assays was at 3.25 pg*ml$^{-1}$. The mean intra-plate CV was 2.87%, while the inter-plate CV were calculated for three different concentrations of the standard curve and were, respectively, 4.73% (2,000 pg*ml$^{-1}$), 0.7% (124 pg*ml$^{-1}$) and 2.77% (7.8 pg*ml$^{-1}$).

### Data analysis

We tested the effect of lutein and cholesterol supplementation on each recorded behavior, on circulating T and on body mass, by using a full factorial general linear mixed model. Treatment (carotenoid treated—Caro; cholesterol treated—Chol; controls—C) and time (day 0, day 15 and day 30) were included as fixed factors, while a random intercept was specified for individual. Significant differences between and within groups were ascertained by pairwise differences of least square means (Student's *t*-test) expressed by confidence intervals (reported in square brackets). After checking the normality of residuals and homogeneity of variance, only song duration (square root) and T (log) were transformed, but original values are reported in Fig. 2. All statistical analyses were performed with SAS 9.3 (SAS Institute Inc., Cary, NC).

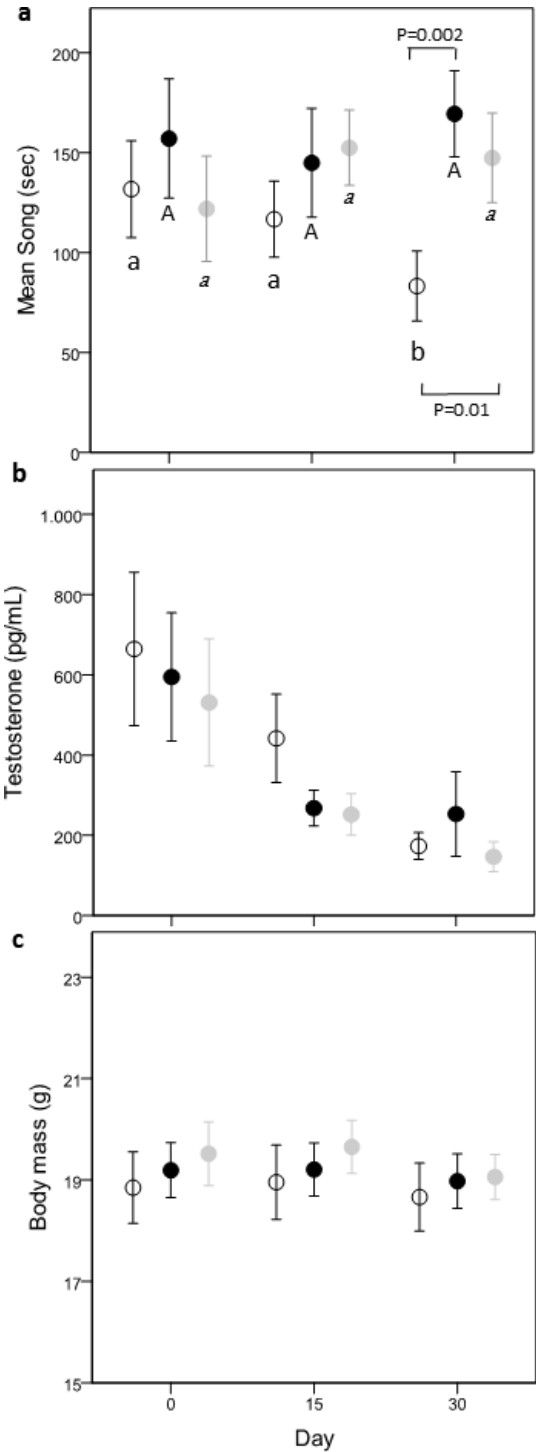

**Figure 2** **Variation of time spent singing (A), testosterone (B) and body mass (C) in the three treatments: Controls (open circle), Carotenoids (black circle) and Cholesterols (grey circle).** Different letters refer to post-hoc within-group significant differences between days in relation to day 0 (Controls: small case, Carotenoids: uppercase, Cholesterols: italic). *P* values refer to significant difference of post-hoc comparisons between dietary groups, within the same day. Dots are mean values and bars s.e.

## RESULTS

### Singing activity

There was a nearly significant interaction effect between treatment and time (time × treatment, $F_{(4,80)} = 2.32, p = 0.06$; treatment, $F_{(2,80)} = 1.64, p = 0.20$; time, $F_{(2,80)} = 0.11, p = 0.89$). Since singing activity can be affected by the quantity of food eaten, we controlled for this variable introducing eating behavior as covariate. The new model showed a significant time × treatment effect (time × treatment, $F_{(4,80)} = 2.64, p = 0.04$; treatment, $F_{(2,80)} = 2.22, p = 0.12$; time, $F_{(2,80)} = 0.49, p = 0.62$) and a significant effect of the total time spent eating behavior ($F_{(1,119)} = 9.54$, $p = 0.0025$, ß$= -0.0079 \pm 0.003$). Comparisons within groups showed that C-males decreased their singing activity with time (day 0–day 30; $-2.76 \pm 1.20$ [$-5.15, -0.36$]; day 15–day 30: $-2.66 \pm 1.22$ [$-5.08, -0.24$]), while this was not the case in Caro- and Chol-males (post-hoc results shown in Fig. 2A). Both Chol- and Caro-males sang significantly more than C-males during day 30 (Caro- vs. C-males: $-4.60 \pm 1.49$ [$-7.55, -1.64$]; Chol- vs. C-males: $-3.46 \pm 1.46$, [$-6.36, -0.56$]), while on day 0 singing activity did no differ between the three treatment groups (Fig. 2A).

### Testosterone

Circulating T concentrations decreased over time ($F_{(2,80)} = 28.80, p < 0.0001$) in all groups independently from food treatments (time × treatment, $F_{(4,80)} = 0.34, p = 0.85$; treatment, $F_{(2,80)} = 0.64, p = 0.53$; Fig. 2B) reaching very low values on day 30 (about 200 μg/ml). Males of the three treatments did not differ in T levels on day 0.

### Body mass

Treatment and time had no effect on body mass (time × treatment, $F_{(4,80)} = 0.28, p = 0.89$; treatment, $F_{(2,80)} = 0.27, p = 0.77$; time, $F_{(2,80)} = 2.89, p = 0.06$) (Fig. 2C). Body mass did not differ among the 3 treatment groups before the start of the food supplementation (all $p > 0.43$).

### Other activities

The time spent eating did not vary between groups (time × treatment, $F_{(4,80)} = 0.75, p = 0.56$; treatment, $F_{(2,80)} = 1.03, p = 0.36$; time, $F_{(2,80)} = 2.12, p = 0.13$), neither did the duration of resting behavior (time, $F_{(2,80)} = 2.65, p = 0.08$, (time × treatment, $F_{(4,80)} = 0.66, p = 0.62$; treatment, $F_{(2,80)} = 0.26, p = 0.77$). Birds did not change their locomotor activity during the experiment (time × treatment, $F_{(4,80)} = 1.45, p = 0.23$; treatment, $F_{(2,80)} = 1.51, p = 0.23$, time, $F_{(2,80)} = 2.41, p = 0.10$).

The total time spent on self-maintenance activities increased over time, irrespectively of the treatment (time, $F_{(2,80)} = 20.35, p < 0.0001$; day 0: $56.53 \pm 9.54$ s, day 15: $139.51 \pm 17.79$ s, day 30: $190.89 \pm 18.02$ s; time × treatment, $F_{(4,80)} = 0.99, p = 0.42$). Although birds did not differ in the duration of self-maintenance activities on day 0 (all $p > 0.38$), there was an effect of treatment ($F_{(2,80)} = 4.49, p = 0.01$; C: $88.07 \pm 8.79$ s, Caro: $163.06 \pm 23.12$ s, Chol: $137.33 \pm 16.57$ s), as overall, Caro birds spent more time than C birds in performing this activity ($74.99 \pm 25.46$ [$23.54, 126.44$]), while any difference emerged between Chol- and C-males ($49.26 \pm 25.03$ [$-1.32, 99.85$]) or Caro- and Chol-males ($25.73 \pm 25.03$ [$-24.86, 76.31$]).

## DISCUSSION

Our study showed that male zebra finches exposed to a decreasing short photoperiod down-regulated both their song activity and T level. This decreased song activity was not observed in males that received extra lutein or cholesterol, with males of both groups maintaining a constant song activity compared to the pre-treatment condition, although T levels were decreasing. As expected, in the treated birds circulating T decreased with the decreasing photoperiod but it was unrelated to undirected song. Overall, the results show for the first time that zebra finch song is influenced by food quality independently from caloric content.

Our results showed that birds eating *ab libitum* food but without any dietary enrichment decrease their song activity in parallel with the decrease of circulating testosterone, but birds treated with lutein or cholesterol did not. Generally, the reproductive state of an individual and the associated hormonal profiles are important for the expression of birdsong. In several species the neural system of birdsong is modulated by T (*Nottebohm, 2005*), but undirected song can be unrelated to reproduction and be T-independent (*Bernard & Ball, 1997*; *Eens, 1997*; *Pinxten et al., 2002*; *Riters et al., 2002*). For example, male European starlings perform undirected song year-round, including periods when T is very low (*Eens, 1997*; *Pinxten et al., 2002*; *Casagrande et al., 2014*). This does not preclude that undirected song can be regulated by T in certain contexts. For example, starlings treated with exogenous T outside of the breeding season sing more frequently than non-treated individuals (*Van Hout, Eens & Pinxten, 2011*), indicating that T modulates this complex signal.

In our study, control birds decreased their song activity with decreasing circulating T but they never ceased singing, even when the levels of this hormone were very low. One possibility is that the low levels of T measured on day 30 were sufficient to ensure the activation of the neuronal patterns involved in song performance (*Pröve, 1974*). Similar to our results, Perfito and colleagues (*2008*) found that males kept under a short photoperiod with regressed gonads, low levels of luteinizing hormone, and full access to food sang more than birds exposed to a long photoperiod and fully activated reproductive state, indicating that food access, more than reproductive state, determines song activity in this species. In our study, males were not acoustically isolated from each other. We used this protocol because this species is highly gregarious and since they could not interact physically with each other, acoustic contact allowed them to continue to interact socially, which is important in this species. Nevertheless, we are confident that housing conditions did not affect the results in a significant way, since we have been able to detect between-group differences. We consider it unlikely that the positive effect of our dietary treatments was mediated by an increase in energy uptake as both nutrients are calorie-free, which is supported by the absence of an effect on body mass in our experiment. We think that it is likely that both cholesterol and lutein could have improved the health status of the birds and that the upregulation of singing activity paralleled this, as was also suggested in a recent review on the relationship between birdsong and oxidative stress (*Casagrande, Pinxten & Eens, 2016*). It has been previously observed that a lutein-rich diet affects the concentration of reactive oxygen metabolites and increase the undirected song rate of male starlings

(*Casagrande et al., 2014*) kept in outdoor aviaries and exposed to a short and decreasing photoperiod identical to the artificially one created in the present study. In line with this is another study on the same species showing that male song can convey information about the oxidative status of the individual (*Costantini et al., 2015*). In the present study we could not determine the oxidative status of birds, because the small size of the species did not allow us to collect enough blood volume to assess both hormone concentrations and oxidative status. In contrast to the results obtained in male starlings, where the dietary effect on song expression was already present after one week, the effect in zebra finches was present only after one month. This difference in timing could be due to differences in the housing conditions between the two studies, which were in large outdoor aviaries for starlings and in individual indoor cages for zebra finches. Starlings had been exposed to the lutein treatment in winter, when the ambient temperature was very low and when an amelioration of their condition with the consumption of antioxidants would have a greater impact in comparison with the situation of a constant environmental temperature of 20 °C, as was the case for zebra finches. Indeed, we observed an increase of oxidative damage during the experiment in starlings, supporting this possibility. However, additional studies are needed to further support this explanation.

## CONCLUSIONS

None of the behaviors measured during the present study, except song activity, were significantly affected by the treatments, showing that birdsong is more sensitive to an improvement of the nutritional or physiological conditions than are other behavioral traits (*Cotton, Fowler & Pomiankowski, 2004*; *Johnstone, Rands & Evans, 2009*). Our data here suggest that the condition of the signaler, here related to the physiological condition of the individual (*sensu Hill, 2011*), can be affected by the quality of the environment and in particular the nutritional value of food ingested, and that song activity can therefore potentially deliver information about the quality of the territory or about the foraging skills of the signaler. It has been demonstrated that singing activity is very sensitive to food availability (e.g., *Ritschard & Brumm, 2012*), but studies showing the effect of food quality independent of caloric content are only starting to emerge (*Van Hout, Eens & Pinxten, 2011*; *Casagrande et al., 2014*). This is particularly important for species that sing year round, for which seasonal changes in food quality can be a potential limiting factor in their singing performance. For example, carotenoids are produced mostly by photosynthetic organisms that usually show marked fluctuations between vegetative and non-vegetative phases. The availability of micronutrients like carotenoids can indeed significantly fluctuate during time and space (*Eeva et al., 2011*). Eeva and colleagues found that in deciduous or mixed forests eggs laid by the pied flycatchers *Ficedula hypoleuca* before leaf unfolding of birch had a lower carotenoid concentration than the ones laid later, indicating an important temporal pattern in dietary carotenoid availability. Different biomes can extensively differ in their capacity to provide carotenoids (*Eeva et al., 2011*). Thus, it is not unlikely to assume that singing activity can be affected by all these environmental parameters and future studies should address this possibility in free-living birds.

## ACKNOWLEDGEMENTS

The authors are very grateful to Michaela Hau and Sue Anne Zollinger for revising a previous version of the manuscript and for providing highly valuable comments. We thank Peter Scheys and Geert Eens for their assistance in many steps of the experiment and for solving several logistic issues. We thank Pralle Kriengwatana and one anonymous reviewer for raising very constructive comments in the revision process, contributing to improve the quality of the manuscript.

### Funding

SC was supported by a FWO post-doctoral Pegasus Marie Curie Fellowship (grant 05-05-1.2.205.13N). RP and ME were supported by the University of Antwerp (TOPBOF) and FWO-Flanders. EZ was supported by an Erasmus Placement program. The funders had no role in study design, data collection and analysis, decision to publish, or preparation of the manuscript.

### Grant Disclosures

The following grant information was disclosed by the authors:
FWO post-doctoral Pegasus Marie Curie Fellowship: 05-05-1.2.205.13N.
University of Antwerp (TOPBOF).
FWO-Flanders.
Erasmus Placement program.

### Competing Interests

The authors declare there are no competing interests.

### Author Contributions

- Stefania Casagrande conceived and designed the experiments, performed the experiments, analyzed the data, contributed reagents/materials/analysis tools, wrote the paper, prepared figures and/or tables.
- Rianne Pinxten and Marcel Eens conceived and designed the experiments, contributed reagents/materials/analysis tools, reviewed drafts of the paper.
- Erika Zaid performed the experiments, reviewed drafts of the paper.

### Animal Ethics

The following information was supplied relating to ethical approvals (i.e., approving body and any reference numbers):

The study was approved by the ethical committee for animal experiments (ECD) of the University of Antwerp (ID number: 2014-21).

### Data Availability

Edmond Repository, Max Planck Digital Library: http://edmond.mpdl.mpg.de/imeji/collection/ldlgaaT46nzyL2YW/item/LBmJNlR0zmBVTX_r.

## Supplemental Information

Supplemental information for this article can be found online at http://dx.doi.org/10.7717/peerj.2512#supplemental-information.

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
