# Peer review of "Positive effect of dietary lutein and cholesterol on the undirected song activity of an opportunistic breeder"

_PeerJ, doi:10.7717/peerj.2512_

## Round 0.1 · original submission · Major Revisions

Your manuscript received very constructive comments from two reviewers and they are both in broad agreement about the quality and presentation of your work. Whilst the study will be a useful contribution to the literature in this area, there are currently a number of problems with the manuscript that need to be addressed so that readers can better understand exactly what your methods are and how they should best be interpreted. I encourage you to engage very thoroughly with all of the comments raised by the reviewers, when revising your paper.

·

Basic reporting

Overall this manuscript was well written with appropriate background literature that is properly cited. Figures are also clear. Please see comments in “General comments” section for a few minor corrections.

Experimental design

The design of the experiment is suitable for the research question but there are a few things that need to be clarified.

-line 167: Are these levels of lutein what zebra finches would get in the wild?

line 174: Were the authors able to determine how much seed/water was eaten, to quantify approximately how much each bird was dosed with?

-line 176: Clarify that “seeds provided to birds were cholesterol free” means seeds provided to birds in the lutein and control groups.

-line 182: The description of the procedures are confusing because they jump back and forth in time. I would strongly suggest adding a timeline showing times at which treatment, mass, behaviour, and blood samplings occurred (as well as light:dark hours at the different sampling points). This would be very helpful.

-line 186: “analyzed for 20 minutes of recording” sounds strange. Perhaps “analyzed in 20 minute recordings”?

-line 192: are beak rubbings the same as beak wiping?

- line 194-196: Is there any way to dissociate directed and undirected songs, since birds were housed in one room? For example, were they able to see each other/other (female) birds? How close were cages? Providing more details about the housing conditions would probably answer many of these questions. Please also describe how singing behaviour was scored.

-line 197: Does day 0 correspond to the first day of treatment? Or one day before treatment?

Validity of the findings

My main concerns regard the authors’ interpretation that song is the only behaviour that affected by their treatments, even though 1) the effects on song seem to be derived from posthoc tests on nonsignificant main effects and interactions (see comments about line 246 and lines 272 below) and hence do not seem very robust; 2) self-maintenance behaviours were significantly affected by treatment (but ignored by authors in discussion, see comments about line 261). I also believe the authors make rather strong statements about what their results mean (e.g. comments about lines 277, 278-280) that are not fully supported by their findings, or at least require much more work in order to prove. I also had some queries about their raw data file that I believe need to be addressed.

- Line 221: I wonder if separate GLMMs for each behaviour is the best way to approach this analysis, especially because some of these behaviours may be highly correlated (e.g. resting and locomotor activities). Moreover, singing and testosterone may be interdependent, so testosterone could be included as a covariate in song rate analyses.

- Line 246: So there was no significant effect of treatment on song rate or song rate x time? Can the authors please justify why they do posthoc analyses despite the nonsignificant effect? I think it is crucial to clarify this if the authors want to make strong claims that song is affected by their treatments.

- Line 250: Why did it take 30 days to see a difference in song rate? In studies of food quantity and song rate, these effects are evident much sooner. A study cited by the authors also finds effects of carotenoid supplementation after after 3-7 days (Van Hout et al. 2011). What might be causing the difference between studies? If it takes a month for the relationship between food quality and song to manifest, then is it a reliable signal for receivers?

- line 257: How did duration of resting behaviour increase but locomotor activity stay constant?

- line 261: The authors find that treatment affects self-maintenance activities, but no posthoc was conducted. Did Caro groups have the highest self-maintenance behaviours compared to the other groups? I think more detailed analyses of this effect is warranted.

- line 272: I think the statement/conclusion that “none of the other measured behaviours were affected by the treatment” is disingenuous, since the authors report a significant effect of treatment on self-maintenance behaviours.

- line 277: “Song is influenced by food quality independently from content of calories.” What exactly does this mean? Did the authors mean that song is influenced by food quality and not food quantity? I would ask authors to provide some evidence of quantity of food consumed (i.e. same across all groups) to verify this statement. Perhaps carotenoid and/or cholesterol-supplemented birds sang more and ate more, but stored extra food consumed as body fat with no change in body mass (see Kriengwatana & MacDouall-Shackleton 2015 Physiol Behav 88,208-215 for indications that zebra finches can increase body fat without increasing body mass).

- line 278-280: The authors may be over reaching with their statement that their results show “food quality can overrule any potential hormonal constraints in this species, showing further evidence of the independence of undirected song performance on sex steroids”. This is because the authors only measured circulating testosterone (no other sex steroids important for song/singing behaviour such as estradiol), so it is premature to conclude based on this study that food quality can overrule hormonal constraints in zebra finches. Moreover, there are dissociations between circulating levels of sex steroids and locally-synthesized levels of sex steroids in the brain – that is, behaviours can be maintained without detectable levels of plasma sex steroids. For instance, Western song sparrows maintain high singing rates and territorial behaviours year round despite low levels of circulating testosterone, but these behaviours are maintained by conversion of DHEA into estradiol in the brain (e.g. see Soma et al. 2002 Horm Behav 41, 203-212). I think these possibilities need to be mentioned and discussed with regards to how undirected song could be independent from sex steroids.

- line 307-310: same as above comments about lines 278-280.

- line 314: I would like to see a more detailed explanation of why housing conditions did not affect results.

- line 318: Please elaborate. Improved health status in what sense?

-line 342-344: Given that body mass (and perhaps fat?) was not affected by your treatments, isn’t it misleading to say that your data show that the “condition” of the signaler is determined by quality of nutrition?

- lines 346: It would be good to interpret your results in light if the decreasing photoperiod used in the experiment. Would birds with better quality nutrition also sing more during breeding season?

- Raw data file: Authors indicated that they coded behaviours based on 20 min recordings, which should be 1200 seconds if the time across their 5 measures (eating, resting, sqrtsong, selfmaint, locomotor) is summed. However, there seems to be quite a bit of variation between birds. For example, sum ZF03 = 500 seconds; ZF07 =750 seconds; ZF45 = 300 seconds (approximately). Can the authors please explain why the sum of times of observed behaviours varies so much between individuals?

Additional comments

Overall I think this is a nice study that addresses two important issues: what effect food quality has on song and what is signalled in undirected song. As the authors point out, both these issues are relatively neglected in the literature. I would like to see the results published, but nonetheless have some major concerns (outlined in the “validity of findings” section) that I believe the authors must first address.

Below are some minor comments:

-line 59: “sing at higher rates”

-line 64: Lynn et al. 2010 found this in what species?

-line 80-82: Please clarify: does this refer to all species or just zebra finches? Same goes for lines 95-98, i.e. sex steroids, reproductive state, androgens, please clarify which species these results apply to.

line 93: “presence duration of the females” sounds strange

line 99: Please explain: What does lutein do? Why is it an important micronutrient? Why might it have helped EUST cope with inflammatory response/antioxidant? I only realized later in the methods that lutein is probably a carotinoid. It would be very helpful if the authors could describe lutein and its functions in more detail here.

-line 111: “acquired through diet”

-line 112: It’s not immediately clear to me how cholesterol is “calorie-free”. Can the authors please explain?

- line 120: At this point I’m still unclear on why lutein and cholesterol would affect song rate, especially since the authors describe singing as being energetically demanding, yet expect calorie-free molecules to change song rate. I think an explicit hypothesis here would help to clarify things. Keeping this hypothesis with the expected relationships from line 141 here would help as well.

-line 124: Bringing this up here is a bit odd. I would describe reasons for using this study species before arriving at the aim of the study.

-line 124-126: this sentence is quite awkward and difficult to understand, please re-phrase.

- line 141: Please explain abbreviation of T before using it. Also, the authors switch between “T” and “testosterone” frequently throughout the manuscript. I would prefer it if the authors chose to stick with one or the other.

-line 144: Still, after reading the introduction I’m not convinced there is a good reason to believe that micronutrients (that don’t provide energy) would affect undirected song rate. I think the authors need to really emphasize their reasons earlier in the introduction.

-line 163 -165: Delete “since” or “thus” from this sentence.

-line 201: No lutein results reported.

- Line 223: It needs to be made clear in the introduction that lutein is a carotinoid, otherwise this label makes no sense. Authors could also consider calling this group “lutein treated”.

- Line 233, 239, 245: I’m a bit confused with the authors’ reporting style, as it appears that they begin with reporting the posthoc results before reporting the main or interaction effects from the LMM. I would suggest reporting things in the reverse order.

- Line 239: If authors have body fat data that would be even stronger support for no effect on condition, since it seems that zebra finches can regulate body fat and body mass independently (e.g. Kriengwatana & MacDouall-Shackleton 2015 reference).

- lines 328-336: I think this paragraph can be omitted. It doesn’t seem to add anything to the discussion (relevant information has already been given in introduction).

Reviewer 2 ·

Basic reporting

The manuscript reports a study testing the effects of calorie-free food supplementation on song rate in zebra finches. Overall the topic of the study is interesting and most previous studies on nutrition and song rate in songbirds have used food manipulations differing in calories. Yet, the manuscript is somewhat difficult to read as it combines many different things and not all of the detailed information is really relevant for the main question., All the physiology and photoperiod data do not really link to the main question on song rate so that the manuscript in part reads as if several things are combined posthoc in a manuscript.

The main finding is a weak effect of food supplementation on song rate as only the control group is not changing, yet the manuscript is difficult to evaluate as no information is given on how song rate was actually measured. Also, information is missing on the distribution of the 20min analysis blocks across treatments. As all birds were in the same room, they will affect each others’ singing and thus at certain times more may sing than others (see comments below). The methods are just too brief with respect to how the key data were collected and analyzed.

While the results of song output are given in seconds, it is not clear if these are the sums of motif durations, or if silent intervals between song motives are included or if these measures are derived in a different way. Moreover, zebra finches sing in bouts, so are individuals who, say sing 60 sec in one bout, treated in the same way as those who sing in many smaller bouts? Do individuals with more seconds singing also sing more songs/motifs? How was Observer software used to measure singing duration? Does Observer calculate spectrograms or was the whole acoustic analysis done by ear?

Some more specific comments:

L21 the aim should be to answer a question (the investigation itself usually is a not a scientific aim)

L22 there is mixed evidence that song rate is sexually selected. This statement here (and elsewhere is a bit too simplistic). Surely in in zebra finches, as the evidence of song rate as being relevant in females choice is very mixed (see Riebel 2009 for a discussion on this)

L28/29 this sentence combines two very different results with each other and the first and second half to not link well with “ while”

L46, song can be an expression of quality but the statement here is too simplistic as this is not always the case…

L49 most studies show that singing has very low energetic costs; the statements here and later in the manuscript (l331ff) are misleading and in part wrong

L71 I found this section very “unusual”. The distinction in between directed and undirected song is made almost exclusively in zebra finches and to generalize this across species seems strange, surely with this terminology. In any case, the distinction seems completely unnecessary as the whole experiment is about undirected songs (see l 194), so this section and unusual general distinction is not relevant for a reader.

L151 and 154, line 151 states that birds were in individual cages but l154 mentioned aviaries. Please clarify how birds were kept. If there were in cages, could the birds see and hear each other?

L166 the control males are not mentioned in this section. Did all birds receive food/water at the same time or were experimental cages supplemented separately form the control cages, and thus were approached by caretakers more often? How were song recordings made relative to feeding times?

L182 were all birds recorded simultaneously (this is what I assumed first but somehow it seems that there were three different recordings days (l182/183). How were recordings session/days distributed across birds and treatments. Given that birds were in the same room, they will have affected each other’s singing activity so it is crucial to know when males of the different treatments were recorded/analyzed relative to each other.
Only video recordings are mentioned and no sound recordings. Was sound taken from video files? Was it always clear that the focal male and not a neigbour was singing? How was song analyzed in seconds without special sound analysis software?

L186 was the start time of the time segments different for the different treatments? Randomly assigning the time slots could lead to strong differences in when the 20 min segment started relative to video onset. It also is not clear if all recordings for all male were done on the same day and simultaneously.

L191 again, song rate is the key variable of the manuscript but it is not mentioned how it was measured. Rate is usually songs (motifs) per minute but the results are not given as rate but as singing duration

L312 this information was not mentioned in the methods

Experimental design

see above

Validity of the findings

see above

---

## Round 0.2 · Minor Revisions

You have made good revisions in line with the reviewers comments and your manuscript will be acceptable once the final relatively minor edits (as suggested by the reviewer, below), are made.

·

Basic reporting

No comments

Experimental design

No comments

Validity of the findings

No comments

Additional comments

The authors have submitted a nicely revised manuscript that I am happy to endorse, pending a few more suggestions:

- The manuscript reads alright, but would benefit greatly from editing/proof-reading by a native English speaker.
- line 80: Jesse & Riebel 2012 did not show that directed and undirected song differ in "genetic control", but rather cite studies such as Jarvis et al. 1998 which show that immediate early gene expression differs. I think in this situation gene expression is used as a proxy for neural activity, so I suggest the authors revise this sentence and update their citation.
- It would be useful to give the reader an idea of what "other behaviours" in the data file refer to.

---

## Round 0.3 · accepted · Accept

Thanks for making these final revisions. The manuscript now looks good.